# *OpenAgents*: An Open Platform for Language Agents in the Wild

**Tianbao Xie**[*♠◇] **Fan Zhou**[*♠◇] **Zhoujun Cheng**[*♠◇] **Peng Shi**[*◇] **Luoxuan Weng**[*◇] **Yitao Liu**[*♠◇]
**Toh Jing Hua**[◇] **Junning Zhao**[◇] **Qian Liu**[♡◇] **Che Liu**[◇] **Leo Z. Liu**[◇] **Yiheng Xu**[♠◇] **Hongjin Su**[♠◇]
**Dongchan Shin**[♠◇] **Caiming Xiong**[♠] **Tao Yu**[♠◇]

[♠]The University of Hong Kong [◇]XLang Lab [♡]Sea AI Lab [♣]Salesforce Research
{tbxie,taoyds}@cs.hku.hk, koala99.zf@gmail.com, z6cheng@ucsd.edu

## Abstract

Language agents show potential in being capable of utilizing natural language for varied and intricate tasks in diverse environments, particularly when built upon large language models (LLMs). Current language agent frameworks aim to facilitate the construction of proof-of-concept language agents while neglecting the non-expert user access to agents and paying little attention to application-level designs. We present *OpenAgents*, an open platform for using and hosting language agents in the wild of everyday life. *OpenAgents* includes three agents: (1) Data Agent for data analysis with Python/SQL and data tools; (2) Plugins Agent with 200+ daily API tools; (3) Web Agent for autonomous web browsing. *OpenAgents* enables general users to interact with agent functionalities through a web user interface optimized for swift responses and common failures while offering developers and researchers a seamless deployment experience on local setups, providing a foundation for crafting innovative language agents and facilitating real-world evaluations. We elucidate the challenges and opportunities and set a foundation for future research and development of real-world language agents.

## 1 Introduction

Intelligent agents are broadly conceptualized as autonomous problem solvers with the ability to sense their environment, decide, and act upon that environment (Wooldridge & Jennings, 1995; Sutton & Barto, 2005; Russell, 2010). With the advent of large language models (LLMs) (Brown et al., 2020; Chen et al., 2021; Chowdhery et al., 2022; OpenAI, 2023b; Touvron et al., 2023), recent implementations from the academic, industry, and open-source communities have leveraged this concept to create language agents. These agents are capable of utilizing natural language to perform a variety of intricate tasks in diverse environments, showcasing notable potentials (Yao et al., 2022b; Chase, 2022; Gravitas, 2023; OpenAI, 2023a; Wang et al., 2023a).

Meanwhile, current agent frameworks such as LangChain (Chase, 2022), AutoGPT (Gravitas, 2023), Gentopia (Xu et al., 2023a), BabyAGI (Nakajima, 2023), AgentVerse (Chen et al., 2023), Agents (Zhou et al., 2023c) provide proof-of-concept implementations and console interfaces largely tailored for developers. This often restricts access to a wider audience, particularly those not versed in programming or consoles. Current agent benchmarks are constructed within specific environments for deterministic evaluation, especially in scenarios involving coding (Yang et al., 2023a), tool utilization (Li et al., 2023c; Patil et al., 2023; Qin et al., 2023b), web browsing (Shi et al., 2017; Yao et al., 2022a; Deng et al., 2023; Zhou et al., 2023b) or a combination of above (Liu et al., 2023). These offer initiatives toward agent evaluations while general agent usage is bound to be connected to more open or unlimited environments and real users with their special needs and materials.

---

[*]Equal contribution.

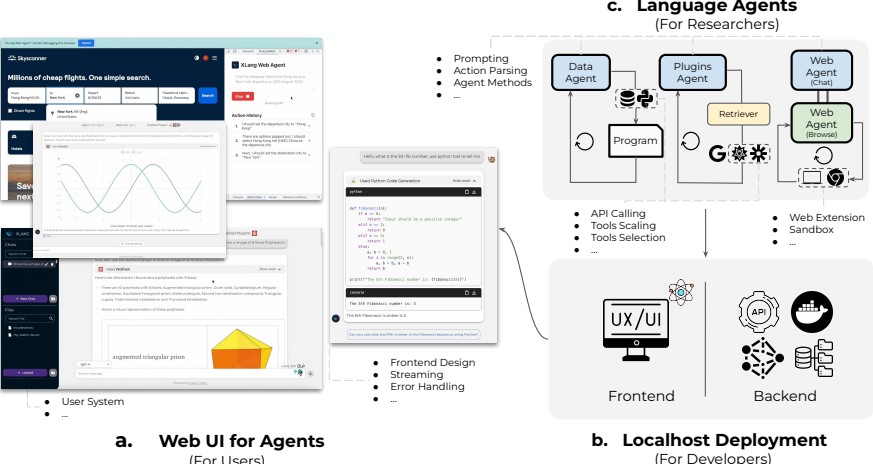

Figure 1: The *OpenAgents* platform for general users, developers, and researchers. (a) General users can interact with the agents via our online web interface, instead of programmer-oriented console or packages. (b) Developers can effortlessly deploy the frontend and backend for further developments given our codes. (c) Researchers can build new language agents or agent-related methods given the examples and shared components, and see how they perform with the web UI. Our *OpenAgents* serves to be a simple and versatile platform for using, developing, and evaluating language agents.

Aiming to develop LLM-powered offerings for a broader user base, OpenAI has crafted and deployed well-designed products[1], specifically Advanced Data Analysis (previously known as Code Interpreter), Plugins, and Browse with ⓑ Bing, leveraging their further trained models (which has been reverse engineered and accepted by many (Zheng et al., 2023b; Gou et al., 2023)), business logic code, and a nurtured software community (e.g., OpenAI plugins store). Despite the significant success attained, the models and business logic code have not been open-sourced due to business considerations, hindering not only users from free access but also developers and researchers from further exploring, evaluating, and improving upon them.

Recognizing this, *OpenAgents* stems from the motivation to democratize agent access as an open-source platform for using and hosting agents, currently encompassing three integral agents: Data Agent for data analysis with Python and SQL, Plugins Agent for 200+ tool usages, and Web Agent for autonomous web browsing. We believe that for LLMs to reach their full potential, they must transition from purely theoretical or developer-centric tools to dynamic, interactive systems that cater to a diverse user base. As depicted in Figure 1, general users can readily explore agent capabilities via the online Web UI without any coding expertise required. In addition, *OpenAgents* provides full business logic and research codes for developers to easily deploy it locally and researchers to further introspect and build language agents. Lastly, given all provided above, *OpenAgents* is meant to be a realistic and holistic human-in-the-loop agent evaluation platform: out of real needs, real users interact with agents to fulfill their tasks, and the whole human-agent interaction traces and user feedback are recorded for further evaluation. Compared to existing benchmarks and platforms, *OpenAgents* provides an in-the-wild environment where agents tackle a variety of genuine user needs.

During building *OpenAgents*, we first underscore the significance of effectively specifying application requirements via LLM prompting, a process that often requires crafting instructions that cater to backend logic, enhance output aesthetics, and safeguard against adversarial inputs. Our findings indicate that the build-up of such instructions can, at times, be substantial, posing challenges in terms of token limitations and context handling for the LLMs. Additionally, for effective real-world deployment, agent models must not only

---

[1]https://chat.openai.com

exhibit high performance but also be able to handle real-time, interactive scenarios, such as streaming, to provide an optimal user experience. Furthermore, our exploration reveals that current research often gravitates towards idealized performance metrics, sometimes sidelining critical real-world considerations, such as the trade-offs between system responsiveness and accuracy, and the nuanced complexities introduced when application-based failures arise, potentially obfuscating the true capabilities of the LLMs.

For future directions and extensions based on *OpenAgents*, we envision the expansion of new agents, agent-related methods, models, and tools for researchers. Plus, the built-in web UI paves the way towards human-in-the-loop agent and LLM evaluation and interaction in the wild under realistic needs, which could benefit the NLP and HCI communities. Finally, we hope that our codebase, which includes off-the-shelf frontend and backend codes and shared components of agents, will inspire the development of more innovative applications.

## 2  Related Works and Preliminaries

Building agents that operate intelligently in specific environments has a long history across the fields of traditional artificial intelligence (Kaelbling et al., 1987; Maes, 1990; Brooks, 1991; Russell, 2010), reinforcement learning (Sutton & Barto, 2005; Mnih et al., 2013; Finn et al., 2017), and recent language agents (Yao et al., 2022b; Yang et al., 2023a; Wang et al., 2023a) which are built on LLMs.

Generally, language agents can be formalized as a partially observable Markov decision process (POMDP) $(\mathcal{H}, \mathcal{S}, \mathcal{A}, \mathcal{O}, \mathcal{T}, \mathcal{R})$ with natural language space $\mathcal{H}$, state space $\mathcal{S}$, action space $\mathcal{A}$, observation space $\mathcal{O}$, transition function $\mathcal{T} : \mathcal{S} \times \mathcal{A} \rightarrow \mathcal{S}$, and reward function $\mathcal{R} : \mathcal{S} \times \mathcal{A} \rightarrow [0, 1]$. Given the dialogue history (user queries and agent responses) $h \in \mathcal{H}$, an agent generates executable action $a_t \in \mathcal{A}$ and interacts with the environment. An action is executable if it is within the action space $\mathcal{A}$ and incurs a change in the state $s_{t+1} \in \mathcal{S}$, and an execution feedback as observation $o_{t+1} \in \mathcal{O}$. The interaction loop repeats until certain stop tokens as action is generated. It is worth noting that currently most LLM-based language agents typically do not require training, and as a result, rewards are often not regarded.

Table 1: Comparison between *OpenAgents* and current existing works on building prototypes and benchmarks on agent concept. Online stands for whether it is deployable and can be online-hosted. UI stands for providing a user interface. #Tools is the number of tools contained. Feedback stands for supporting feedback from users. Web. is short for web browsing. "Controlled" means the framework/benchmark sets up deterministic environments ahead for the agent to operate actions, while "Wild" ones let the agent take actions in the open-ended real-world environment. The "*" marker indicates their agent only implements basic web browsing methods (e.g. web crawl) or uses external APIs (e.g., google search). The "+" marker indicates *OpenAgents* offers additional features beyond just statistical numbers, for example automatic selection on tools. All statistics in the table are collected by September 2023.

| Name | Interface | | | Environment | | |
| --- | --- | --- | --- | --- | --- | --- |
| | Online | Feedback | UI | Coding Env. | #Tools | Web. |
| AutoGPT (Gravitas, 2023) | ✗ | ✓ | CLI | Wild | 15 | ✓ |
| BMTools (Qin et al., 2023a) | ✗ | ✗ | - | Controlled | 32 | ✗* |
| BabyAGI (Nakajima, 2023) | ✗ | ✗ | - | Controlled | - | ✗* |
| Gentopia (Xu et al., 2023a) | ✗ | ✓ | CLI | Controlled | 15 | ✗* |
| Open Interpreter (Lucas, 2023) | ✗ | ✓ | CLI | Wild | 1 | ✗ |
| GAs (Park et al., 2023) | ✗ | ✗ | Web | - | - | ✗ |
| AgentVerse (Chen et al., 2023) | ✗ | ✗ | Web | - | - | ✗ |
| Camel (Li et al., 2023b) | ✓ | ✗ | Web | - | - | ✗ |
| Agents (Zhou et al., 2023c) | ✓ | ✓ | Web | Wild | 11 | ✗* |
| AgentScope (Gao et al., 2024) | ✓ | ✓ | Web | Wild | - | ✗* |
| *OpenAgents* (ours) | ✓ | ✓ | Web | **Controlled & Wild** | **≥200$^+$** | ✓$^+$ |
| ChatGPT Plus (closed-source) | ✓ | ✓ | Web | **Controlled & Wild** | **≥500** | ✓ |

Numerous frameworks and codebases surrounding language agents have been established. One category comprises the construction of some proof-of-concept, catchy prototype implementations (Chase, 2022; Gravitas, 2023; Nakajima, 2023). The core technique is prompting LLMs and demonstrating astonishing results in certain cases. On this basis, some improved and updated conceptual frameworks (Xu et al., 2023a; Zhou et al., 2023c) have been proposed. Another category emphasizes the evaluation of language agents. These evaluations are often conducted in fully simulated environments (self-hosted web pages (Yao et al., 2022a; Zhou et al., 2023b), etc.), semi-real environments (for instance, utilizing real tool API calls (Qin et al., 2023b)) or combination (Liu et al., 2023; Wang et al., 2023b). Additionally, there are frameworks focusing on the study of multi-agents and their behaviors (Li et al., 2023b; Park et al., 2023; Chen et al., 2023; Hong et al., 2023; Wu et al., 2023). Table 1 shows an overall comparison of these frameworks and codebases.

# 3 Platform Design and Implementation

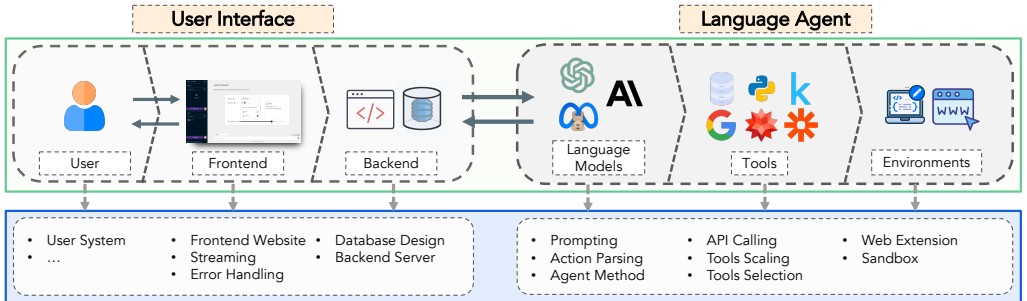

Figure 2: System overview of *OpenAgents*' architecture comprising (1) **User Interface**, a bridge that facilitates communication between the user and the agent, and manages backend operations; and (2) **Language Agent**, encompassing the language model, tools, and environment, driving the agent's decision-making processes. The flow of interaction typically follows from the user needs, through the user interface, and culminates in the language agent taking specific actions using its various components.

## 3.1 Systematic Design

In this section, we introduce our systematic efforts in *OpenAgents*, aiming towards addressing this gap between existing agent frameworks and real-world functional agent applications.

The *OpenAgents* architecture can be generally split into two parts corresponding to Figure 2: (1) User Interface, including both the frontend and backend; (2) Language Agent, including language models, tools, and environments. *OpenAgents* provides a well-built interface for user-agent communication. Upon receiving instructions from the users, the agent will then plan, and take action in the environment via using tools.

**User Interface** We have devoted significant effort to crafting a user-friendly and highly functional User Interface. We have tackled a multitude of highly reusable business logic to use and host agents. This effort has resulted in robust support for various technical aspects within the demo, such as backend server operations, error handling, data streaming, etc. Our goal has been to make the *OpenAgents* not only user-friendly but also to provide enhanced usability. Although these aspects were not the central focus of the current agent frameworks (Chase, 2022; Xu et al., 2023a; Zhou et al., 2023c), we firmly believe that these behind-the-scenes efforts should not be. They hold significant meaning for future research, contributing to the overall robustness and usability of the system.

**Language Agent** In our design, the language agent comprises three essential components: the language model, the tool interface, and the environment. Currently, we adopt our prompting method based on the approach introduced in ReAct (Yao et al., 2022b). The agent generally follows a sequential process of Observation → Deliberation → Action in each

turn of interaction. We also prompt the language model to produce easily parsable text. The tool interface includes parsers capable of translating this text into executable actions, such as generating code or making API calls. Subsequently, these actions are executed within the corresponding environment. We've also invested considerable effort into behind-the-scenes challenges, like stable API calling, tools scaling, and building a sandbox environment.

## 3.2 Practical Implementation Challenges

The implementation journey unveiled several challenges essential for in-the-wild agent readiness. Below are the simplified insights and highlights into the core facets of our implementation. For more details, please kindly refer to Appendix A.

> For **User Interface** implementation, we tackle the following challenges:

1. *Adaptive Data Mapping* §A.1.1 Drawing from database terminology, we employ the concept of `DataModel`. This model effectively converts various raw data types—such as text, codes, images, and tables—into formats optimized for both human-side pretty formatting, LLM contexts, and persistent data storage, streamlining communication between system components and external entities.

2. *Strategic Data Storage* §A.1.2 We adopt a strategic data storage paradigm catering to the multi-user nature of *OpenAgents*. Utilizing in-memory storage for temporary variables, Redis (Carlson, 2013) for global variables, and MongoDB (Banker et al., 2016) for user-centric data orchestrated an efficient data management and retrieval ecosystem.

3. *User-Centric Interface* §A.1.3 An Adaptive User Interface was crafted, bridging the interaction chasm between users and the system. This UI, catering to various operational environments, renders rich media and interactive content (images, code snippets, console outputs, and interactive visualizations, etc.), augmenting user engagement and task efficiency.

4. *Real-time Response Streaming* §A.1.4 A streaming approach was adopted to mitigate the latency inherent in receiving long text completions. Based on streaming API and push-down automata, this innovation allowed for real-time parsing and rendering of generated tokens, significantly ameliorating the immediacy of user feedback.

5. *System Robustness* §A.1.5 Enhancing agent robustness was identified as a quintessential requirement for delivering a realistic user experience. Key areas including effective failure handling (e.g., API calling failure), prompt response generation (e.g., LLM calling APIs pool set and streaming handling), and token overflow management (chat history over length for LLM to handle) were delineated and addressed, ensuring reliable functionality under diverse real-world scenarios.

6. *Browser Control via Chrome Extension* §A.1.6 A Chrome extension was engineered, endowing our web agent with direct browser control capabilities. This setup facilitates real-time user monitoring and intervention during web interactions, and enables web browsing commands on the user side, enhancing user control and trust in the system.

> For **Language Agents**, we provide solutions to the following challenges:

1. *Data Grounding* §A.2.1: Our system lets users upload files which agents can process by writing codes. We've thus established a grounding source pool to store user-uploaded data. With `DataModel`, each file type is linearized and indexed by its name, allowing agents to retrieve and use content as instructed by humans.

2. *Automatic Tool Selection* §A.2.2 Users previously had to select a plugin manually, like OpenAI Plugins (OpenAI, 2023a), for command execution. Recognizing the challenge of manual plugin selection from numerous options, we've integrated a feature namely "Auto Selection", which auto-detects the most relevant tool based on user instructions, streamlining the process.

3. *Automatic Tool Scaling* §A.2.3: While tool creation poses unique challenges, especially for agent use within the LLM infrastructure, we've sourced API provider information from platforms like RapidAPI and OpenAI plugins store. This approach has led to over 200 high-quality plugins, albeit with challenges that occasionally require human oversight. Further exploration is needed for efficient scaling.

4. *Executable Environments* §A.2.4: By "executable", we imply the transformation of language model outputs into actionable tasks within a specific context. Constructing an application-level, multi-user executable environment poses challenges due to the need for safety, robustness, and functionality. We managed to implement sandbox environments catering to code execution, API interactions, and web navigation. These environments serve as a comprehensive testbed for agents, facilitating tasks like code generation, plugin interactions, and web manipulations.

Collectively, these advancements contribute towards architecting a more coherent, robust, and user-friendly platform. Moreover, We believe they also provide a rich repository of learnings, promising to guide future implementations in similar domains.

## 4 *OpenAgents*

In *OpenAgents*, we develop three distinct agents: namely the **Data Agent** for data analysis, the **Plugins Agent** for plugin integration, and the **Web Agent** for autonomous web browsing. The three agents are experts in different domains, just like OpenAI's ChatGPT Plugins (OpenAI, 2023a). Moreover, our implementation is purely based on the top of open language APIs (and theoretically any other open-sourced LLMs), such as GPT-4 (OpenAI, 2023b), and Claude (Anthropic, 2023). We hope *OpenAgents* can serve as a pioneer in the move towards democratizing potent, real-world language agents. Further, we contend that *OpenAgents* can serve as an experimental testbed for researchers across various disciplines who share an interest in the future trajectory of language agents.

### 4.1 Data Agent

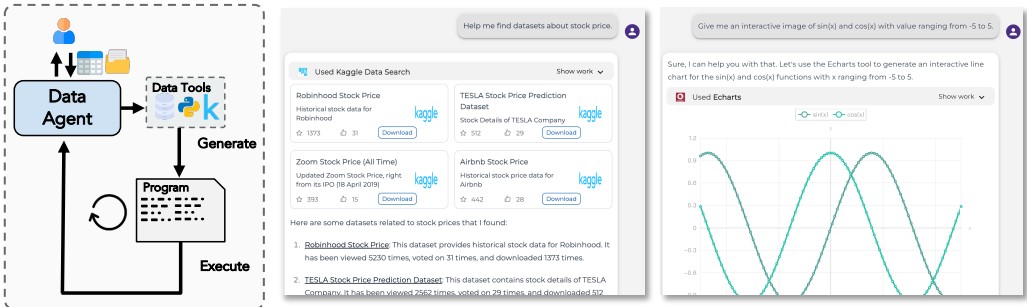

Figure 3: Pipeline (left) and demonstrations (mid and right) of Data Agent.

The data agent has been designed and implemented to deal with a wide spectrum of data-related tasks that real users encounter daily. We support code generation and execution in two programming languages: Python and SQL. We also integrate several data tools for the agent to use: (1) Kaggle Data Search (datasets search on Kaggle via calling API); (2) Data Profiling (heuristic data profiling providing basic data information); (3) ECharts Tool (interactive ECharts plotting). We prompt the agent to proactively use these data tools to respond to user requests. Recognizing the intensive coding requirements for data agent, we have opted to embed language models in the tool, and let the tools generate code rather than the agent. Specifically, tools such as Python, SQL, and ECharts will generate code, thereby harnessing the language models' full programming prowess and alleviating the strain on the agent itself.

Equipped with these data tools, the agent adeptly manages various data-centric requests. It transcends the boundaries of mere text and code generation to proficiently perform

data queries, visualization, manipulation tasks, etc. An illustrative example is depicted in Figure 3, where users can upload custom files, such as tables and images, and make successive inquiries about the uploaded data. While we've highlighted the data agent's key capabilities, the full potential of our data agent extends beyond this brief overview. Much like OpenAI's Advanced Data Analysis(previously known as Code Interpreter) (OpenAI, 2023a), we hope the data agent can serve as a versatile tool for myriad user workflows; for more use cases, please see Appendix B.1.

## 4.2 Plugins Agent

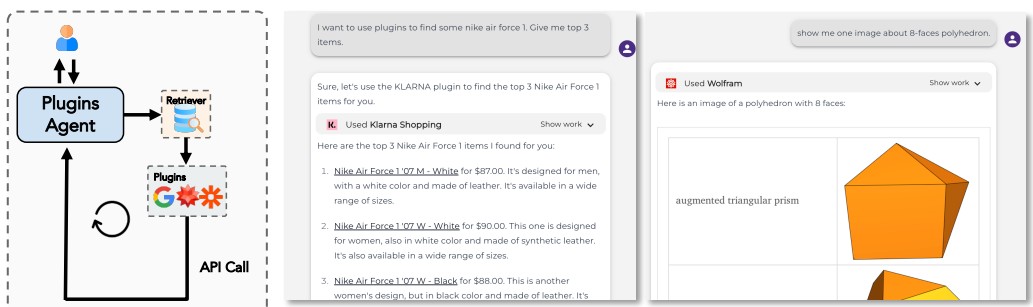

Figure 4: Pipeline (left) and demonstrations (mid and right) of Plugins Agent.

The plugin agent has been meticulously designed to cater to the multifaceted requirements of users' daily tasks which necessitate additional plugins, such as shopping, searching, news reading, and website creation. We have integrated over 200 plugins from various sources, including prominent plugins like (1) Google Search; (2) Wolfram Alpha; (3) Zapier; (4) Klarna; (5) Coursera; (6) Show Me; (7) Speak; (8) AskYourPDF; (9) BizToc; and (10) Klook. Special attention has been paid to ping the APIs, function calling interface, and API response length, enabling LLM-based agents to optimally leverage plugins.

Users can choose one or multiple plugins they would like to let their agent leverage based on the given instructions of their needs. Examples can be observed in Figure 4. In instances where users are uncertain about the appropriate plugins for their needs, we have incorporated a feature that automatically selects the most relevant plugins based on their instructions. We posit that the plugin agent can be instrumental in myriad scenarios, enriching various aspects of users' daily lives. For a more comprehensive list of use cases, refer to Appendix B.2.

## 4.3 Web Agent

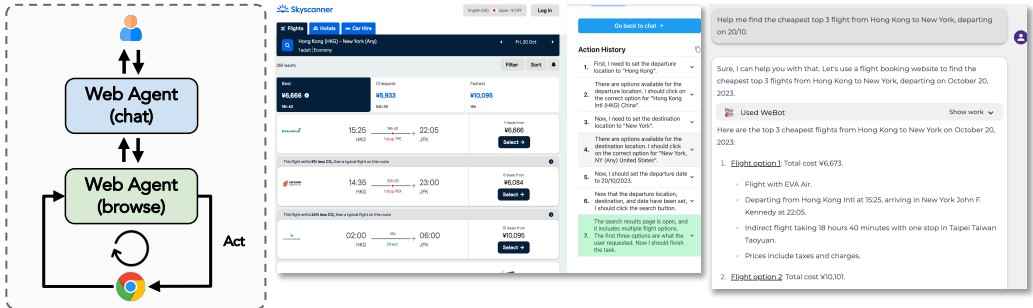

Figure 5: Pipeline (left) and demonstrations (middle and right) of Web Agent.

We present the web agent as a specialized tool designed to enhance the capabilities of the chat agent. The main interaction interface still lies with the chat agent, but when necessary, it seamlessly involves the web agent. The web agent then delivers the final response to

the user, as shown in Figure 5. This design strategy is not just theoretical; it has been implemented and demonstrates several notable advantages.

First, the chat agent systematically processes important parameters like user inquiries or initiating URLs before transferring them to the web agent. This ensures that there is a strong alignment with user intentions and facilitates clearer communication. To clarify further, when dealing with complex tasks that may seem ambiguous or have multiple aspects, the chat agent uses a decompositional approach. Herein, overarching directives from users are segmented into more digestible sub-tasks. The chat agent then interfaces with the web agent using these sub-instructions, sequentially, ensuring more granular and efficient problem resolution. Moreover, our configuration empowers a dynamic interplay of multi-turn web navigation interspersed with chat dialogues, thereby accommodating more layered and adaptable user queries. By distinctly demarcating the roles and responsibilities of the web-browsing agent and the chat agent, we pave the way for the independent evolution and continual refinement of each module. For a deeper dive into practical applications and the potential of the web agent, we invite readers to peruse Appendix B.3.

## 5  From Research to Real-world Deployment

The section reflects on the crucial challenges encountered, the lessons imbibed from real-world scenarios, and the evaluation complexities that surfaced through this transition.

**Challenges of Transforming LLMs into Real-world Apps through Prompting**   When building applications for real users based on LLM prompting, we specify certain requirements through the instructions in the prompt. Some of these instructions are designed to ensure that the output of the LLMs conforms to a specific format for our backend logic to process (e.g., output in the form of a dictionary with specific keys). Others aim to enhance the aesthetic appeal of the output (e.g., listing items separately whenever possible). Additionally, some instructions serve to defend against potential attacks (e.g., refuse maliciously crafted infinite loops in the program by users). As illustrated in the §C and §E, the accumulation of these instructions often results in several hundred tokens, thereby demanding certain requirements on the instruction tracking ability and supported context length of the LLMs. With inferior LLMs, sometimes the improper output format leads to an unsatisfactory frontend appearance or even an inability to complete the response. A positive sign is that there have been significant improvements in these aspects in current open-source models (*i.a.* Peng et al., 2023a; Tworkowski et al., 2023; Zhou et al., 2023a; Peng et al., 2023b; Xiong et al., 2023; Xu et al., 2023c). Additionally, a greater emphasis is required on the foundational development and research of agent models, and on training dedicated agent models (maybe not LLMs) that are tailored for specific domains and requirements. This approach may turn out to be more effective and controllable than relying on purely prompting a general powerful but fixed model.

**Uncontrollable Real-world Factors**   Upon deployment in the real world, we encountered numerous uncontrollable factors triggered by users, internet infrastructure, business logic, and so forth, that haven't been well-modeled, necessitating a reevaluation and often over-turning of many assumptions and forms adopted in past research. We had to assume the possibility of the server hosting the APIs we call crashing, monitoring and robustly completing user commands during such crashes than those assumed in tool-use studies (i.a. Cheng et al., 2023; Schick et al., 2023; Qin et al., 2023a). There could be instances where users may become dissatisfied during the response generation, intervening and causing the language model to halt generation midway. Unpredictable occurrences like CAPTCHAs popping up or advertisements altering the web page, could introduce a degree of randomness even in relatively stable web structures, which is unconsidered in previous autonomous web browsing works (i.a. Shi et al., 2017; Zhou et al., 2023b). These uncontrollable scenarios beckon a deeper exploration or the proposition of more realistic modeling approaches.

**Extra Metrics from Real-world Scenarios**   Research primarily emphasizes performance metrics, frequently overlooking essential requirements derived from real-world scenarios. An instance we learn during implementation is streaming. It allows users to quickly

perceive system responses instead of waiting for lengthy text to generate. Specially designed prompting makes the format of agent response look prettier. These significantly impact user experience. Existing methods (*i.a.* Yao et al., 2022b; Xu et al., 2023b; Lin et al., 2023) haven't considered much of its effects, resulting in slower response times and bad user experience in practical applications, regardless of their superior performance metrics in accuracy. These findings should be contemplated in subsequent application-driven studies. And we also need to mention the trade-offs between the performance and user experience.

## 6 Discussions and Future Work

**Agent Applications**    *OpenAgents* sets up a whole pipeline of building application-level language agents, thus paving the way for other innovative applications such as customizable dialogue systems, multimodal interaction, and automated workflow integrations for end-users. Each of these applications not only offers unique advantages but also collectively contributes to a richer and more user-centric agent application environment.

**Tool and Component Integration**    *OpenAgents* explores and addresses the fundamental requisites for constructing a practical-level agent application, laying down a robust foundation that allows the community to effortlessly expand horizontally by integrating additional components such as tools (e.g., integrate from more diverse API sources like PublicAPIs[2]), extending more foundation models (e.g., recent large multimodal models (*i.a.* Li et al., 2023a; Yang et al., 2023b)), adapting to new UI designs, etc.

**Human-LM Interaction**    Benefiting from the easiness of building new LLM-based agent applications from our platform, we believe *OpenAgents* can consequently be helpful in building application demos for the studies of Human-Computer Interaction (HCI) researchers to delve into the design of more intuitive and user-friendly interfaces (*i.a.* Suh et al., 2023; Kim et al., 2023; Angert et al., 2023), thereby enhancing user engagement and satisfaction.

**In-the-wild Evaluation of LLMs**    Establishing robust, impartial evaluation methodologies for LLMs is essential for an unbiased assessment of their capabilities and performance. Current agent-around benchmarks (*i.a.* Yang et al., 2023a; Liu et al., 2023; Wang et al., 2023b) evaluate agents through benchmarks with pre-collected data and under controlled environments. While these evaluations are crucial, they often do not fully represent the dynamic challenges encountered in real-world scenarios. Platforms like Vicuna Arena (Zheng et al., 2023a) have taken the lead in evaluating chatbots in the wild, marking a significant step towards more realistic evaluation settings. However, there's room for further development in metrics that cater to the nuances of language agents, especially those that tie language understanding with grounding in real-world contexts. Community contributions in expanding or refining these evaluation metrics and platforms are highly encouraged as they would significantly advance the field, providing more accurate insights into the practical performance and capabilities of LLMs.

## 7 Conclusion

In this work, we introduced *OpenAgents*, an open-source platform, tailored to meld the advancements of LLMs with practical, user-focused applications. *OpenAgents* has built agents for three typical applications: data analysis, tool utilization, and web browsing, demonstrating its practical utility. During the development of *OpenAgents*, we identified numerous challenges that were unique to constructing in-the-wild agents. These challenges highlighted the complexities of transitioning from theoretical designs to fully operational agents that cater to actual user needs and interactions. By providing a holistic, transparent, and deployable platform, we aim not only to make the immense potential of LLMs accessible to a wider audience beyond developers and researchers but also to facilitate grounded research through exposure to real-world scenarios, empowering the community to harness and refine the capabilities of state-of-the-art language agents. We sincerely hope *OpenAgents* can inspire and pave the way for a plethora of applications and platforms, streamlining the symbiotic relationship between humans and intelligent language agents.

---

[2]https://github.com/public-apis/public-apis

## Acknowledgments

We extend our profound gratitude to Google Research, Amazon AWS, and Salesforce Research. The munificent gift funds and computational resources they bestowed upon us have been instrumental in the realization of this project. Their unwavering support has been pivotal in the success of our endeavors. We also wish to express our heartfelt appreciation to our contributors for their relentless commitment and insightful perspectives: Roxy Rong, Yixian Zhang, Haotian Li, Xingbo Wang, Chen Henry Wu, Toh Jing Qiang, Jansen Wong, and Jixuan Chen. Equally, our discourse was enriched by the sagacious discussions and invaluable advice from Yiru Chen, Bohan Zhang, Rui Zhang, Ruiqi Zhong, Michihiro Yasunaga, and Ansong Ni. Their collective wisdom has been an invaluable cornerstone in shaping this research.

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

# A   Implementation Details

## A.1   Implementation Details of User Interface

### A.1.1   Data Model to Map Data

Taking database objects as an example, it is common to encounter structures housing dozens or hundreds of tables with thousands or even millions of records. For humans, this data is often interpreted through organized, scrollable windows or analyzed utilizing data visualization tools such as Tableau[3]. On the other hand, an LLM typically cannot process this vast amount of data end-to-end, necessitating a more selective, formulated presentation, e.g., linearization of the first few rows as string. This discrepancy gives rise to the requirement for a more adaptable method of data storage, transcending a mere string mapping or adherence to specific prompting templates.

To address this, we use *DataModel*, which serves as an encapsulating entity designed to intake specific forms of raw data, be it tabular structures, databases, or images. Its primary function is to process and transform this raw data into various output formats that are compatible and comprehensible to a range of designated receivers, including humans, frontend interfaces, computational systems, and LLMs, among others. We establish a suite of over ten fundamental classes to encapsulate data in various formulations, ensuring that each entity involved receives an angle of data that is both relevant and manageable, enhancing efficiency in demo constructions and LLM prompting processes. Through the implementation of the *DataModel* concept, we anticipate a transformative shift in the way data is handled, fostering more coherent and dynamic interactions between humans and AI.

### A.1.2   Strategic Data Storage

Due to the multi-user nature of the actual application, and the numerous conversation records associated with each user, we store data on different mediums based on varying requirements, instead of simply ignoring them in memory or hacking into some temporary files. Temporary variables such as constructed prompts are stored in computer memory; global variables essential within the application are stored in Redis (Carlson, 2013); while user conversation data and distinct user data are stored in a database system, we utilize MongoDB (Banker et al., 2016).

---

[3]https://www.tableau.com/

### A.1.3 User-Centric Interface

Academic-oriented agents typically employ text-based or command-line interfaces, which can impede user interaction and practical task completion. Hence, we introduce an **Adaptive User Interface** that seamlessly integrates user inquiries, agent actions, and associated outputs into the conventional chat workflow. This interface parses tool usage data from backend outputs and responsively renders rich media. Specifically, we design and implement various interfaces that adapt to each environment.

For Data Agent, we interleave images, code snippets, console outputs, and interactive visualizations with markdown text altogether, echoing the structure of computational notebooks frequently employed in data science tasks (McNutt et al., 2023). This enables users to track data analysis workflows interactively and conversationally with our data agent, providing immediate and responsive feedback, thereby enhancing user engagement and sustaining analytical continuity.

For Plugins Agent, we support both user-selected and automated plugin selection. Given the diverse real-world tasks served by different plugins, we render their outputs accordingly. For example, we leverage *Cards* to hierarchically display relevant images and text, typically used for presenting websites, articles, and online products. Thus, we provide users with an immersive and intuitive task-focused experience, enhancing the efficiency and enjoyment of interaction with our plugin agent.

For Web Agent, we incorporate a different interaction method via a Chrome extension that activates upon detection of potential user intents requiring web browsing by our web agent. This extension operates within the browser sidebar, sequentially inspecting, manipulating, and interpreting automatically opened web pages. Meanwhile, we display the execution plans and steps undertaken by our web agent, enabling users to comprehend and follow the process, and intervene if necessary. This liberates users from the manual and tedious task of inspecting and interacting with web pages by themselves.

### A.1.4 Real-time Response Streaming

By default, when we request a text completion from a provider such as OpenAI, the entire completion is generated before being sent back in a single response. However, when we are generating long completions, waiting for the response can take many seconds. To get responses sooner, we can 'stream' the completion as it's being generated, considering the auto-regressive feature of GPT models. This allows us to start printing or processing the beginning of the completion before the full completion is finished. However, it introduces a significant challenge in delineating the specific roles of each generated token in real-time, i.e. some tokens are plain text that can be directly printed out, some tokens serve as API calls that need to be parsed and then can be printed out, some are special tokens that need to be marked for special handling.

Traditionally, real-time identification of the distinct roles of individual tokens - be they directly displayed responses, undisclosed planning sequences, or internal thought processes (Wei et al., 2022; Yao et al., 2022b) - has not been a focal point of research and development in dialogue systems and existing agent frameworks, such as LangChain (Chase, 2022).

Yet, in practical applications, this facet is crucial in enhancing the user experience. ChatML[4], aiming at facilitating the determination of dialogue termination points, and possibly updating grammar rules to handle complex role dynamics within response characters.

We recognize the correlation between this process and automata theory, specifically drawing parallels with pushdown automata (Autebert et al., 1997). Our approach focuses on the early identification of the roles of characters in the data stream, thus enabling a prompt display to the user. This early interaction not only bridges the gap between generation and user receipt but also fosters a more interactive and engaging user experience.

---

[4]https://github.com/openai/openai-python/blob/main/chatml.md

A number of LLMs service providers or self-hosting open-source frameworks provide streaming APIs, such as OpenAI [5], as well as FastChat [6].

For models solely used for chatting, the parsing is relatively simple. In fact, one might argue that parsing isn't required at all; one can merely render these tokens directly onto the frontend using markdown. OpenAI designed a simple ChatML (Chat Markup Language) [7] for this purpose. However, for language agent applications prompted using LLMs, the situation becomes more intricate. The tokens outputted by the LLM might be used to initiate a tool call, or they might signify the tool itself, or even parameters for the tool. They can mark the beginning of a displayed response or the content of the response itself. The roles are multifaceted. One direct approach involves setting markers for the beginning and end of these roles, and then prompting LLMs to output according to these markers. But this demands that LLMs have a strong ability to track instructions and generalize, especially when dealing with formats they have never encountered before. Hence, we shifted the pressure from generation to parsing, attempting to address this using automata. We discovered a strong correspondence between automata and the role identification of streaming tokens. Each state $s$ in the automaton can be mapped to a role of a token (though the granularity might differ slightly, requiring further code-based adjustments). Each character read results in a state transition described by

$$\delta : Q \times (\Sigma \cup \{\epsilon\}) \times \Gamma \to P(Q \times \Gamma^*)$$

, leading to a new state $s'$. The state $s'$ indicates the current role of the chat.

### A.1.5   System Robustness

The robustness of agents is important when it comes to realistic user experience, including but not limited to failure handling, in-time response, and token overflow. Though previous agents might shed light on some of these aspects, *OpenAgents* offer wrapped-up components or off-the-shelf examples, which developers can follow to build robustly functional agents.

**Failure Handling**   is unpredictable, and may occur when utilizing real-world infrastructure, such as various API services including the LLMs API service and Plugin API services from different providers. Identifying the types of these failures and deciding whether to retry or terminate with an error message is a problem that requires design and resolution. We have addressed this issue wherever there is a potential for such problems to arise.

**In-time Response**   is usually hindered by the LLM call overhead (both decoding time and rate limits) or different plugin APIs' response time. As these practical limits are bound by the service providers, we make some workarounds from the client side to either reduce the response time or improve the user experience. First, since multiple threads may call LLMs in parallel, we collect an LLM key pool (e.g., OpenAI API key) to be queried in turn, alleviating the rate limit burden of each single key. Second, we allow users to stop and retry each generation on will. This is crucial because it gives users the freedom to choose whether they would like to wait for the current response when the already generated tokens are not satisfied (also require the *Streaming* technique introduced later) or there is a block or stuck in remote API calling underneath. Without it, users may end up waiting so long for an unwanted response, if not no response. Third, we set a maximum waiting time for each remote API call, and categorize detailed error information for each failed response, so that users will always know what happened inside the agent and how to interact next.

**Token Overflow**   is a common situation in prompting LLMs where the input and output tokens exceed the max token limits of the LLM. We implement a *MessageDataModel* subclass to explicitly truncate message history tokens and extract the information needed (e.g., actions, tool responses) from history. Concretely, the whole chat history is in the agent

---

[5]https://platform.openai.com/docs/api-reference/chat/create

[6]https://github.com/lm-sys/FastChat

[7]https://github.com/openai/openai-python/blob/main/chatml.md

memory by default, and will be truncated from the beginning by round iteratively until the max token limit is met. Since information in all types is wrapped in *DataModel*, we can seamlessly handle the token overflow by calling the handling function anywhere in the data flow, otherwise developers have to design truncation and parsing APIs according to different data types and modalities.

### A.1.6 Chrome Extension

It is difficult to control the browser on the user side since the website or application does not have enough rights in most cases. Let applications run on the computer terminal (e.g. OpenInterpreter [8]) and open mock browser to navigate on the web is easier to implement. However, this method could face some restrictions and some risks in the long shot. It also limits the development of large-scale web assistant applications for demonstration, verification or other purposes. To build a publicly accessible and user-friendly web agent application for demonstration, the agent should be able to control the user's browser directly and can be interrupted or taken over easily by the user. With this in mind, we use the Chrome extension to implement our web agent. The agent can take actions on the user's browser through Chrome debugger API [9]. We parse the output of the agent to extract the action it takes and then use the sendCommand function in Chrome Debugger API to send the action to take to the browser, and then the browser will execute the action instead of the user. During the execution of Web Agent, the user can keep watch on the thinking and action process all the time and can easily interrupt the execution.

---

[8]https://github.com/KillianLucas/open-interpreter
[9]https://developer.chrome.com/docs/extensions/reference/debugger/

### A.2 Implementation Details of Language Agents

#### A.2.1 Grounding Source

Herein, the grounding source represents the data or files that agents can utilize during interactions, especially structured knowledge form (Xie et al., 2022) like Excel files, CSV files and JSON files, etc. We integrate a file-uploading system that allows users to seamlessly upload their files for the agents to process. Agents have the capability to either directly write code or employ alternative methods to read, modify, and save these files. This enables agents to accomplish specific tasks as humans require, rather than merely generating single actions. Consequently, we have implemented a grounding source pool to store and represent data uploaded by human users. We leverage the `DataModel` to provide a linearization method for each supported file type and use the file name as the key for these files. Agents can access the content corresponding to file names and actively invoke the file content as humans desire.

#### A.2.2 Auto-selection on Plugins

Currently, users need to first select a plugin, like the OpenAI Plugins (OpenAI, 2023a), before the system can execute their commands using these plugins. However, in a real-world scenario, choosing an appropriate plugin from hundreds of options can be a daunting task. Oftentimes, even if users know what they want, they might not know which specific plugin can best assist them. Therefore, we've introduced a special feature. Each time the user provides new instruction, we employ a text embedding technique, specifically Instructor (Su et al., 2023), to identify which tools described match closely to the user's instructions, eliminating the need for manual plugin selection.

#### A.2.3 Auto-scaling Plugins

While tools have been constructed under simpler models (Qian et al., 2023; Cai et al., 2023), developing tools for agent use presents unique challenges, requiring attention to both the agent and the LLM infrastructure. Creating tools from scratch ensures quality but is resource-intensive Qin et al. (2023a). We source API provider information from platforms like RapidAPI [10] and the OpenAI plugins store, and test these providers for accessibility by LLMs, This method has enabled us to introduce over 200 high-quality plugins. However, using LLMs for plugin construction has its challenges and requires human intervention. Future work should investigate more efficient scaling techniques.

#### A.2.4 Executable Environments

**Code Execution Environment** We provide SQL and Python environments for code execution. We extract the SQL query from the agent output, and leverage SQL engine [11] to execute the query. For the Python environment, we largely referred to the official Kaggle Dockerfile [12]; the docker container provides a safe and isolated sandbox for code execution.

**API Calling Environment** We make API calls to adhere to the principles of RESTful design (Fielding, 2000; Song et al., 2023). To make sure we're ready for the different needs of each API provider forehead, we organize API calls into separate functions. Within these functions, we manage specific exceptional conditions, such as context overlength (we leverage `DataModel` to handle that) or particular invocation methods. This strategic encapsulation ensures that the agent can seamlessly operate without being encumbered by vendor-specific anomalies. Subsequently, when the agent elects to utilize a particular plugin, the designated plugin name—along with requisite information like the API documentation (herein referenced as open API file)—is relayed to a dedicated parsing and dispatching

---

[10] https://rapidapi.com/hub

[11] https://www.sqlite.org/

[12] https://github.com/Kaggle/docker-python

module. This module, also powered by prompting LLMs, orchestrates the requisite function call and awaits its resultant output. Given that APIs are scattered across various servers on the internet, and their quality and reliability can be inconsistent, the interfaces may change unpredictably.

**Web Environment**   We also explore the possibility of whether the web agent is capable of manipulating real-world web navigation. We let the language agent control the web browser (Chrome in our case) via the Chrome extension built by us and navigate on any websites accessible by the browser. The extension can control the web page by performing actions such as clicking an element or typing values into an element through Chrome Debugger [13]. In this way, the web agent can act like a real human user on the webpage, and the human can also interact with it anytime and continue to act on the webpage. This environment makes the web navigation more real and visible and provides the platform to build a generalist web assistant in the future. On the contrary, Browse with ᛒ Bing can only act behind the screen and the navigation process actually occurs on the server side rather than the client side, which significantly limits the function of web navigation and can hardly be seen, controlled or taken over by the user. This makes it quite hard to build a web assistant that can help users execute daily tasks on the user's computers, and enable more diverse scenarios for any website in the wild unlike screenshotted or self-hosted ones (Shi et al., 2017; Yao et al., 2022a; Deng et al., 2023; Zhou et al., 2023b). We also believe this feature can be beneficial to further research and development in the automatic web agent area.

---

[13]https://developer.chrome.com/docs/extensions/reference/debugger/

# B Use cases

For quick reference, we demonstrate the use cases of data agent, plugin agent, and web agent in Figure 6, Figure 7, and Figure 8 respectively.

## B.1 Data Agent

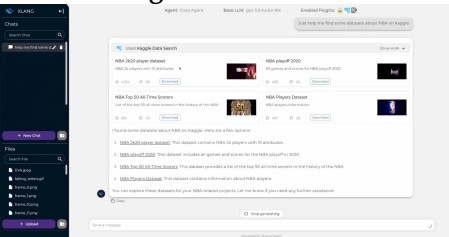

(a) **Kaggle Dataset Search:** Input your subject or domain preference, and the Data Agent will traverse the Kaggle API to identify datasets fitting your specifications.

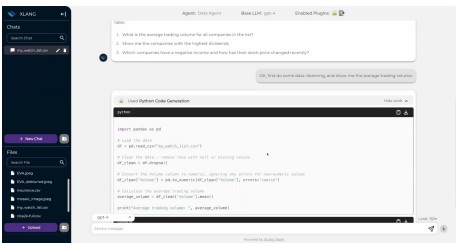

(b) **Python Code Execution:** Utilize the Python tool to craft Python code and execute rudimentary data sanitization on a given table.

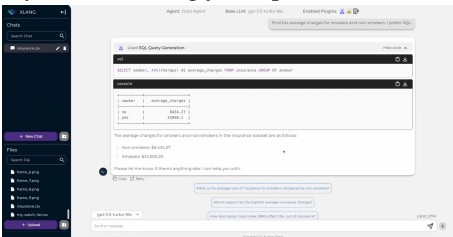

(c) **SQL Query Execution:** Using the SQL tool, the Data Agent can execute data queries on an input table or dataset.

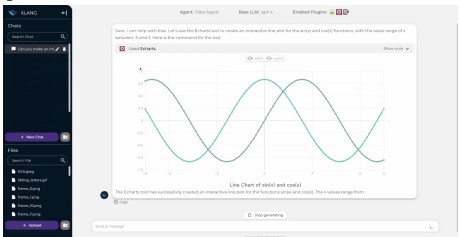

(d) **Interactive Echarts Visualization:** The Data Agent employs the pyecharts package, transforming it into a JSON object. This object can be rendered as an interactive Echarts plot, as facilitated by the ECharts library.

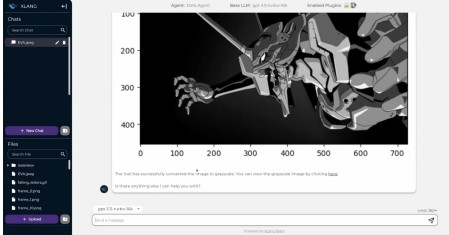

(e) **Image Operations:** Beyond textual and tabular data, the Data Agent can also perform basic manipulations on images.

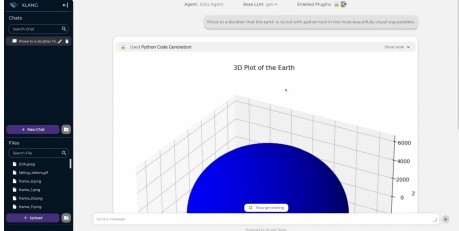

(f) **Interesting Cases:** Demonstrate the earth's roundness, create GIFs with descending letters, or construct a word cloud from an academic article.

Figure 6: Use cases for Data Agent

## B.2 Plugins Agent

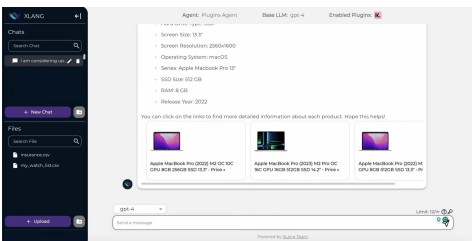

(a) **Shopping with 'Klarna Shopping':** Specify a product and the Plugins Agent, using 'Klarna Shopping', will locate it, presenting it in an aesthetically pleasing layout.

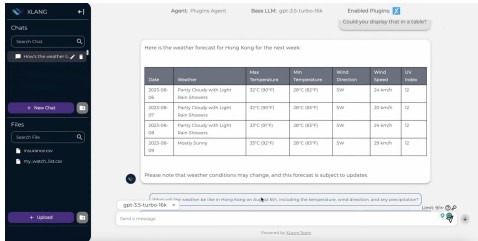

(b) **Weather Updates via 'XWeather':** Seeking weather information? Communicate your location to the Plugins Agent and 'XWeather' will fetch the latest updates.

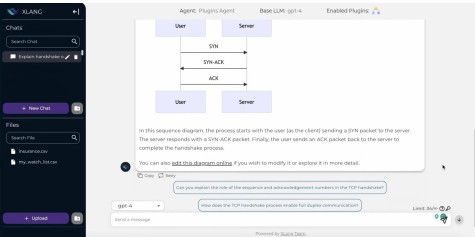

(c) **Concept Visualization with 'Show Me':** Desire a visual representation of an idea? The 'Show Me' plugin, upon receiving your input, will generate and elucidate the concept.

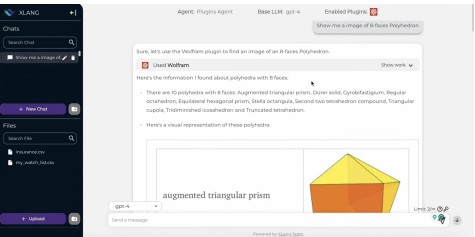

(d) **Inquiry with 'Wolfram Alpha':** Engage with the 'Wolfram' plugin to investigate and gain insights into diverse topics.

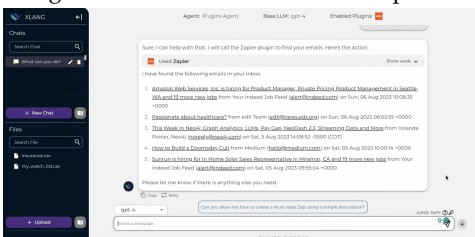

(e) **Workflow Configuration via 'Zapier':** Harness the 'Zapier' plugin to understand and manipulate your Zaps.

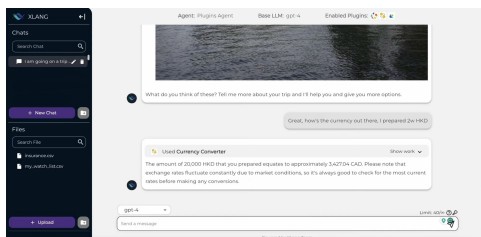

(f) **Travel Planning with Multiple Plugins:** Combine 'Klook', 'Currency converter', and 'WeatherViz' for an enriched travel experience.

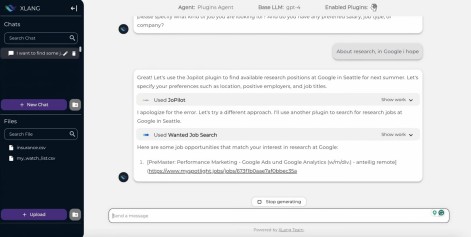

(g) **Automatic Plugin Selection:** Uncertain about plugin functionalities or choices? Convey your requirements to the Plugins Agent, and it will autonomously select the optimal plugin, simplifying your experience.

Figure 7: Use cases for Plugins Agent

## B.3 Web Agent

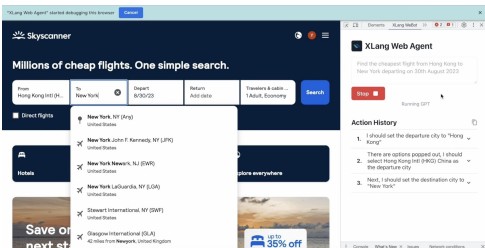

(a) **Flight Finder:** Detail your travel informa­tion, and the Web Agent will scout for flights, for instance, the most economical ones.

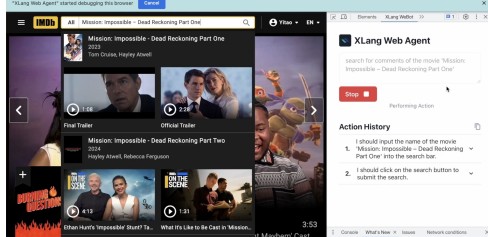

(b) **IMDb Comments Summarization:** Con­sidering a movie? The Web Agent will peruse IMDb comments for your selected film, provid­ing a concise summary.

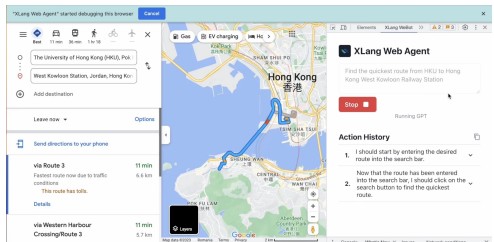

(c) **Google Map Navigation:** Supply your start­ing point and destination to the Web Agent, and it will delineate the route using Google Maps.

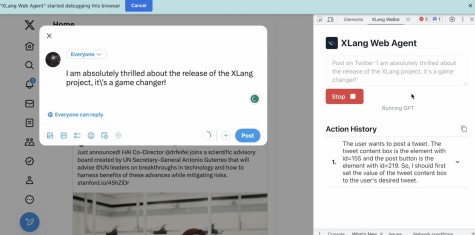

(d) **Twitter Post Creation:** Share your desired topic with the Web Agent, and it will curate and post content to Twitter on your behalf.

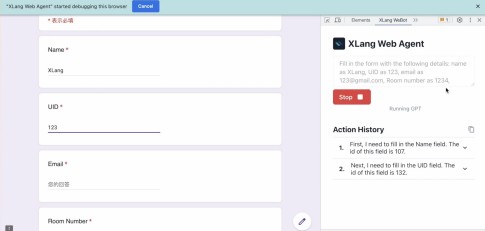

(e) **Google Form Assistance:** For club registra­tions or event sign-ups, provide the necessary form and details to the Web Agent, and it will adeptly complete the submission.

Figure 8: Use cases for Web Agent

# C Key Prompts in *OpenAgents* (Agents)

The main technique behind *OpenAgents* is LLMs prompting, it is crucial for building applications upon LLMs and controls the quality (e.g. retrying after failure), functionalities (e.g. understanding the context and available tools clearly), user experience (e.g. whether the response is in good format) and even safety (e.g. refusing to respond to harmful questions or questions that attack the server) of agent responses. Here we show key prompts we explore and use to prompt LLMs to serve as real-world agents. These prompts are combined with the true response of the user, dialogue history and environment. Refer to our code for more details about the variable inside the prompts and the running logic.

## C.1 Data Agent

**System Prompt**

```
You are an open agent, a friendly and intuitive interface to guide human through
every stage of human data lifecycle. Whether human are loading, processing, or
interpreting data, open agent is always at human's fingertips through our
interactive chat system.

Empowered by an array of innovative tools that can generate and execute code,
open agent delivers robust, reliable answers to human queries. Whenever possible,
 You employs these tools to give human rich insights, like dynamic code
generation & execution and compelling visualizations. And You will always
proactively and correctly using all tools to help with human.

Get ready for a seamless and insightful journey with open agent, the personal
assistant for all things data!

TOOLS
------
You have direct access to following tools.
```

**Format Instructions**

```
RESPONSE FORMAT INSTRUCTIONS
----------------------------

When you use tools or generate final answer, please output a response in one of
two formats:
**Option 1: Explain and Use Tool**
If the response involves using a tool, you can start with a natural language
explanation[Optional], plus exactly one tool calling[MUST]. But **make sure no
any words & answer appended after tool calling json**. The tool calling format
should be a markdown code snippet with the following JSON schema:

```json
{{{{
    "action": string wrapped with \"\", // The action to take. Must be one in the
    list [{tool_names}]
    "action_input": string wrapped with \"\" // Natural language query to be input
     to the action tool.
}}}}
```

[**Restriction**] Please note that ONLY one tool should be used per round, and
you MUST stop generating right after tool calling and make sure no any text
appended after tool calling markdown code snippet. Save your words.
```

```
NEVER EVER EVER make up a tool not in [{tool_names}]
NEVER EVER EVER generate code as action input when using tool. Just input natural
 language by using/paraphrasing human query.

**Option #2:**
Use this if you want to respond directly to the human.
If you want to respond directly to the human without using a tool, provide a
plain natural language response. However, if you initially generated a natural
language response and then decide to use a tool, make sure to include the tool
action and input after the initial response.

Note if the human asks for malicious code, and just respond directly to deny the
request and give your professional reason. Don't use any tool.
The malicious code includes but not limited to:
1. Endless operations and excessive waiting (e.g., while True, long print, input
())
2. System crash (e.g., any risky system command)
3. Data loss (e.g., list or delete files)
4. Leak sensitive information (e.g., os.getenv())
5. Establish network connections (e.g., requests.get())
6. Cause any other security issues

[Mandatory to notice] It is imperative and a must to utilize tools whenever the
human's query tasks that implies using tools, such as searching online,
generating code, executing code, or any other complex functionalities. You must
try to use tools to solve human queries in these cases.

Begin.
```

**Suffix**

```
{input}
```

**Tool Response Template**

```
TOOL RESPONSE:
--------------------
{observation}

THOUGHT
--------------------

Okay, So what's next? Let's assess if the tool response is enough to answer the
human's initial query. Please follow these instructions:

1. Evaluate Tool Response [Mandatory]: Carefully evaluate the tool's response and
 determine if it sufficiently addresses the human's query. Consider the content
and implications of the tool's response.

2. Consider Additional Tool Use [Optional 2 or 3]: If the tool response does not
fully address the query or if an error occurred during execution, you may proceed
 with additional tool usage. However, exercise caution and limit the number of
iterations to a maximum of three. You can start with a natural language
explanation[Optional], plus exactly one tool calling[MUST]. But **make sure no
any words & answer appended after tool calling json**. Follow this format for
additional tool usage:

```json
{{{{
```

```
    "action": string wrapped with \"\", // The action to take. Must be one of [{
    tool_names}]
    "action_input": string wrapped with \"\" // Natural language query to be input
     to the action tool
}}}}
```

[**Restriction**] Please note that only one tool should be used per round, and
you MUST stop generating right after tool calling and make sure no any text
appended after tool calling markdown code snippet.

3. Deliver Comprehensive Answer [Optional 2 or 3]: If the tool response
sufficiently addresses the query, deliver a comprehensive answer to the human.
Focus solely on the content and implications of the tool's response. MUST NOT
include explanations of the tool's functions.

3.1. Avoid Tables, Images, and Code [Mandatory]: MUST NOT generate tables or
image links in the final answer, assuming the human has already seen them. Avoid
generating code in the final answer as well. Instead, paraphrase the code into a
human query if you need to explain it.

Note. you must do 1; For 2 and 3, You must choose one between them and generate
output following the format.

Begin.
```

## C.2 Plugins Agent

**System Prompt**

```
You are open plugins agent, a friendly and intuitive assistant to guide you
through every aspects of your work and your daily life. Open agent is always at
your fingertips through our interactive chat system.

You can aware of what plugins you have, and use the plugins properly in right
order to finish what user wants.

Today is
""".strip() + " "
    + datetime.datetime.now().strftime("%Y-%m-%d")
    + """, and you should adapt the input to fit into the date, for example,
    seasonal information, or today's date as coordinate, etc.

To make your response informative, always speak includes the following
information in MARKDOWN format when responding a message, that is:
1. Natural language explanation, that make explain the API output in a human
readable way;
2. Organized information such as bullet points or MARKDOWN tables, followed by
the links to the items (that in the API output), news etc. if API output contains
 the information;
3. The links should in MARKDOWN format and have value in it. If reference
information is provided in the API output, like links to the items, news etc.
Your explanation MUST provide the links on each items and links can be clicked on
 when API output contains the information. The links better attach on some
natural language explanation through MARKDOWN syntax, for example, - [Renewable
Energy - Center for Climate and Energy Solutions](https://www.c2es.org/content/
renewable-energy/);
4. If there are image we would like to display, please use MARKDOWN syntax to
display it, for example, ;
```

```
5. Try to speak more and show all the information you got in a organized way,
that will make you a better assistant, especially when you are giving the final
answer.

PLUGINS
------
The plugins you can use are:
```

## Format Instructions

```
RESPONSE FORMAT INSTRUCTIONS
----------------------------

When you use tools or generate final answer, please output a response in one of
two formats:
**Option 1: Explain and Use plugin**
If the response involves using a plugin, you can start with a natural language
explanation[Optional], plus exactly one plugin calling[MUST], and ends with no
more words. The plugin calling format should be a markdown code snippet with the
following JSON schema:

```json
{{{{
    "action": string wrapped with \"\", // The action to take. Must be one in the
    list [{tool_names}]
    "action_input": string wrapped with \"\" // Query to be input to the action
    plugin. Pass as much information as possible to the plugin from the history of
     the conversation.
}}}}
```
NEVER EVER EVER make up a plugin not in [{tool_names}]
You MUST pass as much information as possible to the plugin from the history of
the conversation. It could be natural language or structured language like jsonl,
 csv, etc. BUT MUST in a single line.
(Please note that ONLY ONE plugin should be used per response.)

**Option #2: **
If you want to respond directly to the human without using a plugin, provide a
plain natural language response. However, if you initially generated a natural
language response and then decide to use a plugin, make sure to include the
plugin action and input after the initial response.

Begin.
```

## Suffix

```
{input}
```

## Tool Response Template

```
PLUGINS RESPONSE:
--------------------
{observation}

THOUGHT
--------------------
```

```
Okay, So what's next? Are the plugins' response enough to answer human's initial
query? Please follow these instructions:

1. Evaluate plugin Response [Mandatory]: Carefully evaluate the plugin's response
 and determine if it sufficiently addresses the human's query. Consider the
content and implications of the plugin's response.

2. Consider Additional plugin Use [Optional 2 or 3]: If the plugin response does
not fully address the query or if an error occurred during execution, you may
proceed with additional plugin usage. However, exercise caution and limit the
number of iterations to a maximum of 5. You can start with a natural language
explanation[Optional], plus exactly one plugin calling[MUST]. Follow this format
for additional plugin usage:

```json
{{{{
    "action": string wrapped with \"\", // The action to take. Must be one of [{
    tool_names}]
    "action_input": string wrapped with \"\" // Query to be input to the action
    plugin. Pass as much information as possible to the plugin from the history of
     the conversation.
}}}}
```
(Please note that ONLY ONE plugin should be used per response.)

3. Deliver Comprehensive Answer [Optional 2 or 3]: If the plugin response
sufficiently addresses the query, deliver a comprehensive answer to the human.
Focus solely on the content and implications of the plugin's response. MUST NOT
include explanations of the plugin's functions.

Note. you must do 1; For 2 and 3, You must choose one from them.

Begin.
```

## C.3  Web Agent

### System Prompt

```
You are open web agent, a friendly and intuitive assistant to guide you through
every aspects of your work and your daily life. Open agent is always at your
fingertips through our interactive chat system.
Here are detailed instruction for you. Each time you generate response, you
should think step by step to follow instructions below. You are a helpful
assistant that is provided with a plugin called "WeBot" which is a web navigation
 agent tool and should leverage the power of it to help human to fulfill their
needs, such as booking a hotel, buying a ticket, or searching for information,
etc.
Human will ask you questions, and you can use WeBot to help them, they are
assumed to know nothing about the WeBot.
---------------------------
Here are something you MUST remember:
1. After receiving output from the WeBot, you should check
    1.1 whether WeBot was interrupted, if so you should NEVER try again by
    yourself.
    1.2 whether WeBot failed or had error(not because of interruption), if so you
    should tell the human the error.
2. Today is
""".strip() + " "
        + datetime.datetime.now().strftime("%Y-%m-%d")
```

```
         + """, and you should adapt the input to fit into the date, for example,
         seasonal information, or today's date as coordinate, etc.

NEVER EVER EVER use other plugins except WeBot.
TRY YOUR BEST to break the question down into several parts and answer them one
by one.
TRY YOUR BEST to use the WeBot to help you answer the question, you don't need to
 mention that you will use which WeBot, just use it.

To make your response informative, always speak includes the following
information in MARKDOWN format when responding a message, that is:
1. Natural language explanation, that make explain the API output in a human
readable way;
2. Organized information such as bullet points or MARKDOWN tables, followed by
the links to the items (that in the API output), news etc. if API output contains
 the information;
3. The links should in MARKDOWN format and have value in it. If reference
information is provided in the API output, like links to the items, news etc.
Your explanation MUST provide the links on each items and links can be clicked on
 when API output contains the information. The links better attach on some
natural language explanation through MARKDOWN syntax, for example, - [Renewable
Energy - Center for Climate and Energy Solutions](https://www.c2es.org/content/
renewable-energy/);
4. If there are image we would like to display, please use MARKDOWN syntax to
display it, for example, ;

5. Try to speak more and show all the information you got in a organized way,
that will make you a better assistant, especially when you are giving the final
answer.

PLUGINS
------
The plugins you can use are:
```

**Format Instructions**

```
RESPONSE FORMAT INSTRUCTIONS
----------------------------

When you use tools or generate final answer, please output a response in one of
two formats:
**Option 1: Explain and Use WeBot**
If the response involves using a WeBot, you can start with a natural language
explanation[Optional], plus exactly one WeBot calling[MUST], and ends with no
more words. The WeBot calling format should be a markdown code snippet with the
following JSON schema:

```json
{{{{
    "action": string wrapped with \"\", // The action to take. Must be WeBot
    "action_input": string wrapped with \"\" // Natural language query to be input
     to the WeBot.
}}}}
```
NEVER EVER EVER make up a plugin except [{tool_names}]
NEVER EVER EVER generate code as action input when using WeBot. Just input
natural language by using/paraphrasing human query.
(Please note that ONLY ONE WeBot should be used per response.)

**Option #2:**
```

```
If you want to respond directly to the human without using a WeBot, provide a
plain natural language response. However, if you initially generated a natural
language response and then decide to use a WeBot, make sure to include the WeBot
action and input after the initial response.

Begin.
```

**Suffix**

```
{input}
```

**Tool Response Template**

```
PLUGINS RESPONSE:
--------------------
{observation}

THOUGHT
--------------------

Okay, So what's next? Are the WeBot's response enough to answer human's initial
query? Please follow these instructions:

1. Evaluate WeBot Response [Mandatory]: Carefully evaluate the WeBot's response
and determine if it sufficiently addresses the human's query. Consider the
content and implications of the WeBot's response.

```json
{{{{
    "action": string wrapped with \"\", // The action to take. Must be one of
    WeBot
    "action_input": string wrapped with \"\" // Natural language query to be input
     to the WeBot
}}}}
```
(Please note that ONLY ONE WeBot should be used per response.)

3. Deliver Comprehensive Answer [Optional 2 or 3]: If the WeBot response
sufficiently addresses the query, deliver a comprehensive answer to the human.
Focus solely on the content and implications of the WeBot's response. MUST NOT
include explanations of the WeBot's functions.

Note. you must do 1; For 2 and 3, You must choose one from them.

Begin.
```

# D  Key Prompts in *OpenAgents* (Executors)

A number of Executors in *OpenAgents* are implemented by prompting as well. Here we show some of the prompts as exemplars. Due to space limitations, we only show a few typical examples in the article. Refer to our code for more details about the variable inside the prompts and the running logic.

## D.1  SQL Database

**Prompt**

```
Here are chat histories you may refer to, maybe empty.
{chat_history}

Given an input question, first create a syntactically correct {dialect} query to
run, then look at the results of the query and return the answer.
Never query for all the columns from a specific table, only ask for a the few
relevant columns given the question.
Pay attention to use only the column names that you can see in the schema
description. Be careful to not query for columns that do not exist. Also,
remember to wrap the table names in double quotes.
Use the following format:
Question: "Question here"
SQLQuery: "SQL Query to run"
SQLResult: "Result of the SQLQuery"
Answer: "Final answer here"
Only use the tables listed below.
{table_info}
Question: {question}
```

## D.2 API Calling

**System Prompt**

```
You are acting like plugin system that understand user's needs and call APIs
precisely for them.
```

**User Prompt**

```
Here are the endpoints specs:
```
{specs_str}
```
Here is the input string:
```
{input_str}
```
Select the right endpoint called that can process the input string.
You need to wrap the input str into a json object, so that it could be fed into
the function you selected.
During wrapping, you should:
1. modify the value of each key so that it satisfies the requirements in function
 specs.For example, if the type of the value should be a number, then you should
modify it into a number;
2. ignore the information that is not useful or not applicable to the function
you selected.
You fill values into some slots in the input_json, and then call the API. If the
API returns a valid output, then you succeed. Otherwise, you fail.
Return the function called and the json object in the following format:
```
{{
    "endpoint": "xxx",
    "input_json":{{
        "xxx": "xxx",
        "xxx": "xxx",
        ...
    }}
}}
```

```
```
```

**Retry Prompt**

```
Here are the function specs:
```
{specs_str}
```
Here is the input string:
```
{input_str}
```
Select the right function called that can process the input string.
You need to wrap the input str into a json object, so that it could be fed into
the function you selected. During wrapping, you should:
1. modify the value of each key so that it satisfies the requirements in function
 specs.For example, if the type of the value should be a number, then you should
modify it into a number.
2. ignore the information that is not useful or not applicable to the function
you selected.
Return the function called and the json object in the following format:
```
{{
    "endpoint": "xxx",
    "input_json":{{
        "xxx": "xxx",
        "xxx": "xxx",
        ...
    }}
}}
```
You have tried to call function to process the input string but failed. The
output do not have enough information to answer the tool input.
Here is the history of your trials, each element in this list means a trial:
```
{trial_history}
```
You should firstly analyze your trial history, find the value of the key "errors"
 in the output of each trial and check whether there are any errors
Then you may consider changing the input_json or endpoint based on the error
information in your trial history, function specs and the input string.
Return the function called and the json object in the following format:
```
{{
    "endpoint": "xxx",
    "input_json":{{
        "xxx": "xxx",
        "xxx": "xxx",
        ...
    }}
}}
```
```

**Stop Prompt**

```
Here are the function specs:
```
{specs_str}
```
```

```
Here is the input string:
```
{input_str}
```
Here is the output that you get from calling the API:
```
{api_output}
```
You need to decide whether the returned_block contains valid information or not.
Some returned_block may not have enough information to answer the tool input, for
 example, the returned_block may be empty, or return a json that says some kind
of answer.
Answer only by 'yes' or 'no'
```

### D.3 WeBot

**System Prompt**

```
You are a browser automation assistant.

You will be given a user request and DOM of current webpage at a time, you need
to take one action at a time and finally finish the task.

The last page you visited will be further fed into another model who is
responsible for chatting with the user.

You MUST take one of the following actions. NEVER EVER EVER make up actions that
do not exist:

{formattedActions}

You will be be given a task to perform and the current state of the DOM. You will
 also be given previous actions that you have taken. You may retry a failed
action up to one time.

This is an example of an action:

<Thought>I should click the add to cart button</Thought>
<Action>click(223)</Action>

You MUST always include the <Thought> and <Action> open/close tags or else your
response will be marked as invalid.

Rules you MUST follow:
1. If you input something to a search box, YOU MUST FOLLOW:
    1.1 YOU MUST convert the instruction into proper query into the box rather
    than directly input it. e.g. YOU MUST input New York rather than New York
    apartments in the input box of zillow.com when user request about New York
    apartments.
    1.1 If there are some options pop out, you MUST NOT directly go to next action
    . You MUST click one of the options.
2. You must only take one step at a time. You cannot take multiple actions in a
single response.
3. You should check whether your action last time was successful. If not, you
should retry once. If it still fails, you should try another way.
    example 1: The box should be clicked and choose from the options and you just
    setValue and failed, you may consider to use click and then click the option.
    example 2: You click a button once but after checking the page you found that
    the button is not clicked, you should retry once.
```

```
4. You should not consider the action to present the result to the user. You only
 need to do available actions. If info in current page is enough for the user to
solve the problem, you should finish.
5. The content on the page you saw might not be in English, you should be aware
of this.

{plan}

Remember: you do not need to follow this plan exactly, but you MUST follow the
rules above.
YOU MUST MUST check whether there are some options pop out if your last action is
 setValue. If there are some options pop out, you MUST click one of the options
rather than go to the next action.
The id of the elements can be different each time. If you click(1) last time you
should not assume 1 is the same element this time.
```

**User Prompt**

```
The user requests the following task:

{user_query}

{previous_actions_string}

Current time: {current_time}

Current page contents:
{processed_html}
```

**Retry Prompt**

```
The user requests the following task:

{user_query}

{previous_actions_string}

Current time: {current_time}

Current page contents:
{processed_html}

Your last answer has some problem:
{retry_message}
```

# E  User Interaction Statistics

To evaluate the performance and efficiency of our *OpenAgents*, we collected and analyzed a variety of interaction metrics. In total, *OpenAgents* has collected 10K conversations with 56K messages contributed by 7,832 registered users, This section presents a detailed statistical overview of user interactions, including the distribution of conversation durations, the number of conversations initiated, and the frequency and diversity of tool usage during conversations.

## E.1  Conversation Distribution

We analyzed the distribution of conversation lengths across our user base, focusing on the number of turns and tokens. Figure 9 presents these distributions in detail.

Firstly, the average number of turns per conversation is 5.6. While over 35,000 conversations (35%) end within two turns, indicating a single interaction with the agents, the majority of conversations are more extensive, involving multiple-turn engagements with agents.

Secondly, tools play a critical role in agent problem-solving, with an average of 1.95 tool uses per conversation. This includes the automatically triggered *data profiling* tool for uploaded data.

Lastly, the majority of token consumption (over 80%) occurs on the agent side, with the average conversation token length being 1,229.

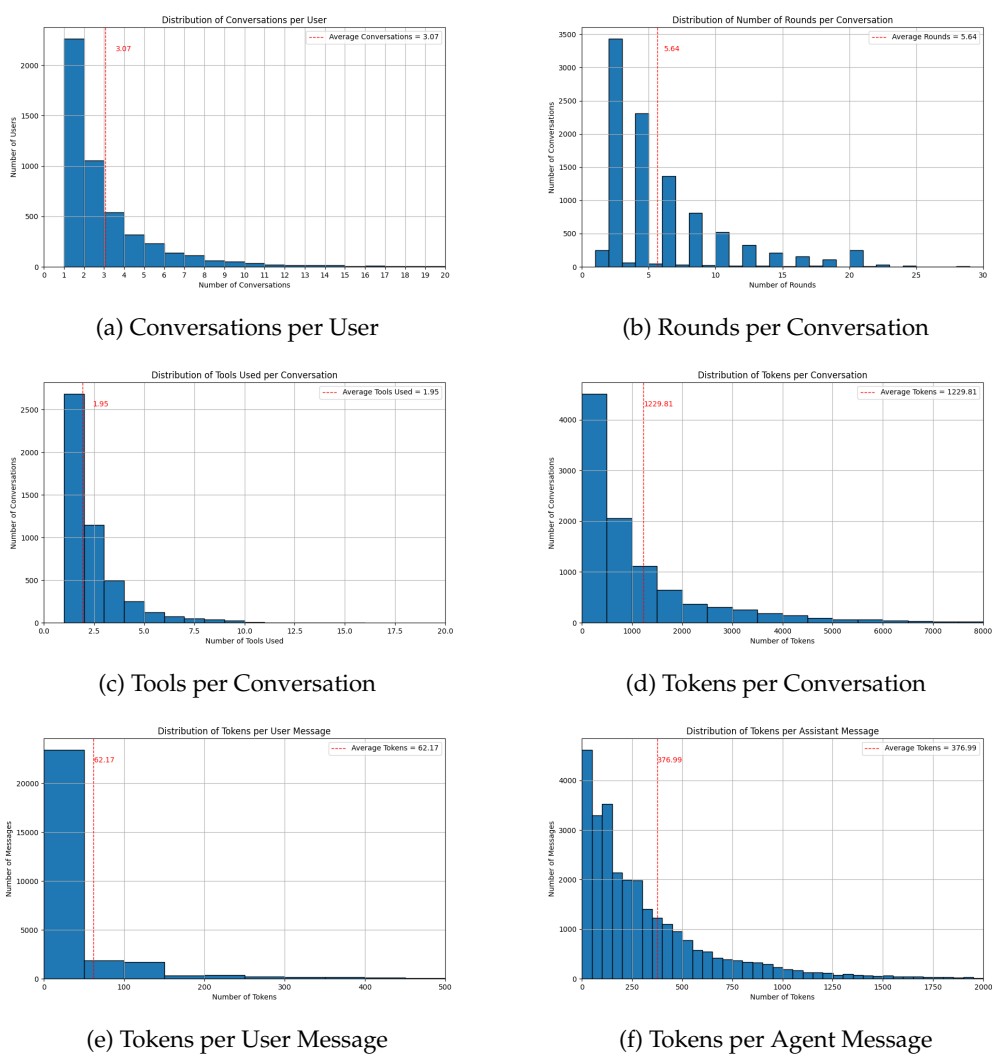

Figure 9: Distribution Statistics of User Interaction Data.

