# OpenReview forum: "OpenAgents: An Open Platform for Language Agents in the Wild"
_colmweb.org/COLM/2024/Conference — COLM_

### Official Review · Reviewer_8ELQ · 2024-05-10

**Rating:** 7
**Confidence:** 4
**Ethics Flag:** 1

**Summary:**

This paper presents OpenAgents, an open-source platform for real-life language agent applications. The authors design three agents: Data Agent, Plugin Agent and Web Agent, and provide interfaces for different types of users (general users, developers, and researchers). The platform's open-source nature may inspire broader community involvement and innovation, advancing the development and application of language agent technologies. Overall, "OpenAgents" represents a high-quality and practically significant work.

**Questions To Authors:**

1. Compared to LangChain, what specific advantages do ‘OpenAgents’ offer? Additionally, can ‘OpenAgents’ be seamlessly integrated with LangChain?
2. Prompting is a critical part of OpenAgents. Does OpenAgents support a diverse set of prompting strategies, or is it easy for users to add their own?

**Reasons To Accept:**

1. This presents an open-source implementation of ChatGPT Plus；
2. This paper provides a well-defined architecture to guide the design of the language agent platform.
3. The proposed platform ‘OpenAgents’ is designed to be scalable and flexible, supporting various research needs and adapting to various computational environments.

**Reasons To Reject:**

1. This paper doesn’t emphasize the difference between ‘OpenAgents’ and ‘ChatGPT Plus’.
2. Most functionality of ‘OpenAgents’ is based on code or APIs, which makes ‘OpenAgents’ like an assistant rather than an intelligent agent. In more complex scenarios, for example, in OS operation and Household tasks, agents need to consider more.

---

> ### Author Rebuttal · Authors · 2024-05-29
>
> Thank you for your positive review and constructive feedback. We appreciate your recognition of the significance and quality of our work on OpenAgents. We have addressed your concerns and provided further clarification on the points you raised.
>
>
> **W1: Emphasizing the difference between OpenAgents and ChatGPT Plus:**
>
> A: Thank you for highlighting this important point. While we don’t have detailed knowledge of ChatGPT Plus’s internals, we emphasize the unique aspects of OpenAgents, including its open-source nature, modular architecture, and wide API and tool integration. Unlike ChatGPT Plus, OpenAgents offers customizable interfaces for different user types and supports the integration of external agents and plugins, enhancing flexibility and extensibility.
>
> **W2: Functionality based on code or APIs:**
>
> A: Thank you for pointing out a common issue with current API-calling-based agents. Although OpenAgents straddles both approaches with our WebAgent, we believe these two methods address different angles. The performance of advanced models like GPT-4 on benchmarks like WebArena and OSWorld is only around 10%, which is not practical for the current users and developers. In comparison, the API-based modeling approach is more easily implemented by current language models, making it a tradeoff in the application.
>
> **Q1: Comparison with LangChain and integration possibilities:**
>
> A: LangChain aims to facilitate the construction of proof-of-concept language agents while neglecting non-expert user access to agents and paying little attention to application-level designs. We have integrated these aspects into the backend logic, frontend logic, and user interface, building a product-grade platform, and uncovered many issues that only emerge in real-world applications, such as the need for handling long text efficiently, and the expansion and maintenance of real tools, etc.
> OpenAgents is designed to be compatible with LangChain. Users can integrate the two platforms easily to further enhance functionality.
>
> **Q2: Prompting strategies in OpenAgents:**
>
> A: Prompting is indeed a critical part of OpenAgents. Our platform supports a diverse set of prompting strategies, including templates, chaining, and context-aware prompting. We have designed OpenAgents to be user-friendly, allowing users to add and customize their own prompting strategies easily. This flexibility ensures that users can tailor the platform to their specific needs and preferences.

---

### Official Review · Reviewer_zoss · 2024-05-11

**Rating:** 5
**Confidence:** 4
**Ethics Flag:** 1

**Summary:**

- This paper introduces OpenAgents, a versatile and open platform designed for the deployment and management of language agents in real-world settings.
- OpenAgents consists of three types of agents: the Data Agent, the Plugins Agent, and the Web Agent.

**Questions To Authors:**

- Do you open-source all 200+ APIs for Plugins Agent?

**Reasons To Accept:**

- The OpenAgents platform is engineered to serve a diverse audience, encompassing general users, developers, and researchers.
- It provides a wide array of features, including a web user interface, streaming capabilities, and more.
- The paper offers an in-depth exploration of agent-based systems, yielding numerous insights that substantially enrich the field.
-  It details the implementation meticulously, ensuring clarity in the processes involved.

**Reasons To Reject:**

- Gathering feedback from actual users is crucial for evaluating the platform's effectiveness and overall user experience. However, the paper does not discuss this aspect.
- The performance of the auto-selection feature for plugins has not been evaluated.
- In terms of agents' design, there are no significant novel contributions.
- It seems that the platform lacks a chat API service capable of handling large volumes of user queries.

---

> ### Author Rebuttal · Authors · 2024-05-29
>
> Thank you for your thoughtful review and for recognizing the broad utility and detailed exploration presented in our OpenAgents platform. We appreciate your constructive and detailed feedback.
>
> **W1: Gathering feedback from actual users**
>
> A: We agree that user feedback is crucial for evaluating the platform's effectiveness. Although our main focus is the design philosophy and technical details of OpenAgents, we discuss the need for new language agent metrics and in-the-wild evaluation in Sec. 5 and 6. Evaluating human-agent interaction is challenging, but we have made significant efforts to improve platform quality. While automatic evaluation isn't reliable yet, we are committed to enhancing OpenAgents based on user feedback.
>
> **W2: Performance of the auto-selection feature for plugins**
>
> A: The goal of the OpenAgents paper is on how features like auto tool selection significantly enhance user experience compared to ChatGPT Plus, rather than examining the capabilities of individual components in detail, which is beyond the scope of this paper.
>
> **W3: Novel contributions in agents' design**
>
> A: While the primary focus of our work is to create an accessible and versatile platform, we also highlight our technical contributions in two key areas.
> 1. **System design:** For Data & Plugin Agents, we follow the ReAct agent definition but introduce techniques such as adaptive data mapping, data grounding, and auto tool selection/tool scaling to transition from proof-of-concept to real-world applications.
> 2. **Web Agent:** We use a Chrome extension for the Web Agent, enabling dialog-based assistance to help users complete tasks through interactive actions.
> We will revise the manuscript to detail our agents' unique design aspects.
>
>
> **W4: Chat API service and handling large volumes of user queries**
>
> A: We appreciate your concern regarding the chat API service. In our released code, we expose the chat/ endpoint API on the server side for handling large concurrent user queries. Users can directly query this API  in their programs/applications for concurrent access instead of on the web demo.
>
>
> **Q1: Missing Plugins**
>
> A: We previously uploaded 16 best-performing tools in our user test as some other public tools suffered from connection or maintenance issues. We have been cleaning and validating the rest tools this month and have uploaded all 200 API tools these days.

---

### Official Review · Reviewer_Cqqw · 2024-05-11

**Rating:** 6
**Confidence:** 4
**Ethics Flag:** 1

**Summary:**

The authors propose an open platform OpenAgents for deploying and interacting with language agents in everyday contexts. It addresses the gap in current frameworks by focusing on accessibility for non-expert users and emphasizing application design. The platform comprises three key agents: a Data Agent for handling data analysis tasks using Python/SQL and other tools, a Plugins Agent equipped with over 200 daily utility APIs, and a Web Agent designed for autonomous web navigation.

**Reasons To Accept:**

1.	Good illustrations of the functional utilities of the proposed framework.
2.	The studied topic is important and appeals to a broad audience.
3.	The provision of an open-source codebase, developed with considerable dedication, is both welcomed and appreciated.

**Reasons To Reject:**

1.	[Writing Style & Technical Depth] The paper presents its content at a too high-level fashion, leaving ambiguities regarding its novelty from a methodological perspective. It fails to clarify the rationale behind the selection and integration of numerous tools, the features to be provided, and what makes the construction of such a platform technically challenging and non-trivial. For instance, the discussion on "tackling the challenge of strategic data storage" in Sec A.1.2 is too brief, offering only two sentences without adequate details. This lack of information does not sufficiently explain the issues related to `data storage` or how the approach is `strategic`.
2.	[No Experiments & Analysis] The paper does not adequately justify the proposed platform due to the absence of quantitative verification. The lack of empirical and numerical support makes it difficult for readers to grasp the effectiveness and operational mechanisms of the platform.
3.	[Preciseness & Timeliness] Assertions and arguments to current works require careful review and updates, given that LLM-Agent is an extremely active and rapidly evolving field of study. For example, in Table 1, some markers and specific numbers are outdated (gathered by September 2023, whereas the submission deadline for COLM is March 29, 2024), potentially misleading new readers and being unfair to the works being compared.
4.	It is recommended to explore additional literature on data agents and multi-agent systems for possible improvements or integration. For instance, a data agent with enhanced data processing capabilities could offer significant benefits [1], and the proposed *three agents* could potentially be enhanced to cooperatively work better  through more customized multi-agent workflows [2].

[1] (arXiv’23, SIGMOD’24) Data-Juicer: A One-Stop Data Processing System for Large Language Models

[2] (arXiv’24) AgentScope: A Flexible yet Robust Multi-Agent Platform

---

> ### Author Rebuttal · Authors · 2024-05-29
>
> Thank you for your thoughtful review and appreciation of our work. We are glad you highlighted the importance of our studied topic, the quality of the illustrations, and the provision of an open-source codebase. We have addressed the specific points you mentioned and will update the paper accordingly if it is accepted.
>
>
> **W1: Writing Style & Technical Depth:**
>
> A: Thank you for highlighting the need for more detailed technical explanations. We have expanded several subsections to provide deeper insights into our method. Specifically, in Section A.1.2, we elaborate on the challenges of strategic data storage, detailing our hierarchical storage strategy (local memory -> Redis -> persistent database) with illustrations. We explain which variables are stored at each level and the benefits of this approach. Additionally, we provide more details on grounding sources in Section A.2.1 and auto plugin selection in Section A.2.2. We will update the paper accordingly if accepted.
>
>
> **W2: No Experiments & Analysis:**
>
> A: We agree that quantitative verification is essential for demonstrating the effectiveness of OpenAgents. In the revised manuscript, we will include human-in-the-loop experiments: 1) user satisfaction surveys, 2) average time to complete selected tasks, and 3) failed case studies. Evaluating human-agent interaction is challenging, but we have made significant efforts to improve platform quality. We have established Discord and Slack channels with over 6,000 users and addressed 87 issues and engaged 16 new contributors. Despite the lack of reliable automatic evaluation, we are committed to enhancing OpenAgents based on user feedback.
>
>
> **W3: Preciseness & Timeliness:**
>
> A: Thank you for the observation. We have updated Table 1 with the latest data as of April 2024 to ensure our comparisons are both fair and current.
> | Name                | Interface                        |          |          |  Environment            |          |     |
> |---------------------|-----------------------------------|----------|----------|-------|-------------------------|----------|
> |                     | Online                            | Human Feedback | UI           | Coding Env.            | #Tools   | Web      |
> | **AgentScope** (Gao et al., 2024)  | ✓ | ✓ | Web | Wild | - | ✗* |
> ...
> We will further review recent literature, including Data-Juicer, and discuss contributions and potential integration points within our platform in the final version.

---

### Decision · Program_Chairs · 2024-07-10

**Decision:**

Accept

**Comment:**

This paper presents OpenAgents, an open-source platform to develop language agents for real-world applications. The supported applications include data agents, plugins agents and web agents.

I believe this open-source platform can be valuable for the community to develop and evaluate LLM-based agents, thus I recommend acceptance. However, the lack of evaluation results is a notable drawback of this work. The authors should revise the paper to include more details of the platform, update the discussion of related work, and add empirical results they promised to add.

[comments from the PCs] Please revise the paper following the AC recommendation, especially the empirical content promised for the next revision.